# NORMALIZED DIRECTION-PRESERVING ADAM

## ABSTRACT

Optimization algorithms for training deep models not only affects the convergence rate and stability of the training process, but are also highly related to the generalization performance of trained models. While adaptive algorithms, such as Adam and RMSprop, have shown better optimization performance than stochastic gradient descent (SGD) in many scenarios, they often lead to worse generalization performance than SGD, when used for training deep neural networks (DNNs). In this work, we identify two problems regarding the direction and step size for updating the weight vectors of hidden units, which may degrade the generalization performance of Adam. As a solution, we propose the normalized direction-preserving Adam (ND-Adam) algorithm, which controls the update direction and step size more precisely, and thus bridges the generalization gap between Adam and SGD. Following a similar rationale, we further improve the generalization performance in classification tasks by regularizing the softmax logits. By bridging the gap between SGD and Adam, we also shed some light on why certain optimization algorithms generalize better than others.

## 1 INTRODUCTION

In contrast with the growing complexity of neural network architectures (Szegedy et al., 2015; He et al., 2016; Hu et al., 2017), the training methods remain relatively simple. Most practical optimization methods for deep neural networks (DNNs) are based on the stochastic gradient descent (SGD) algorithm. However, the learning rate of SGD, as a hyperparameter, is often difficult to tune, since the magnitudes of different parameters can vary widely, and adjustment is required throughout the training process.

To tackle this problem, several adaptive variants of SGD have been developed, including Adagrad (Duchi et al., 2011), Adadelta (Zeiler, 2012), RMSprop (Tieleman & Hinton, 2012), Adam (Kingma & Ba, 2014), etc. These algorithms aim to adapt the learning rate to different parameters automatically, based on the statistics of gradient. Although they usually simplify learning rate settings, and lead to faster convergence, it is observed that their generalization performance tend to be significantly worse than that of SGD in some scenarios (Wilson et al., 2017). This intriguing phenomenon may explain why SGD (possibly with momentum) is still prevalent in training state-of-the-art deep models, especially feedforward DNNs (Szegedy et al., 2015; He et al., 2016; Hu et al., 2017). Furthermore, recent work has shown that DNNs are capable of fitting noise data (Zhang et al., 2017), suggesting that their generalization capabilities are not the mere result of DNNs themselves, but are entwined with optimization (Arpit et al., 2017).

This work aims to bridge the gap between SGD and Adam in terms of the generalization performance. To this end, we identify two problems that may degrade the generalization performance of Adam, and show how these problems are (partially) avoided by using SGD with L2 weight decay. First, the updates of SGD lie in the span of historical gradients, whereas it is not the case for Adam. This difference has been discussed in rather recent literature (Wilson et al., 2017), where the authors show that adaptive methods can find drastically different but worse solutions than SGD. Second, while the magnitudes of Adam parameter updates are invariant to rescaling of the gradient, the effect of the updates on the *same* overall network function still varies with the magnitudes of parameters. As a result, the effective learning rates of weight vectors tend to decrease during training, which leads to sharp local minima that do not generalize well (Hochreiter & Schmidhuber, 1997).

To fix the two problems for Adam, we propose the normalized direction-preserving Adam (ND-Adam) algorithm, which controls the update direction and step size more precisely. We show that

ND-Adam is able to achieve significantly better generalization performance than vanilla Adam, and matches that of SGD in image classification tasks.

We summarize our contributions as follows:

- We observe that the directions of Adam parameter updates are different from that of SGD, i.e., Adam does not preserve the directions of gradients as SGD does. We fix the problem by adapting the learning rate to each weight vector, instead of each individual weight, such that the direction of the gradient is preserved.

- For both Adam and SGD without L2 weight decay, we observe that the magnitude of each vector's direction change depends on its L2-norm. We show that, using SGD with L2 weight decay implicitly normalizes the weight vectors, and thus remove the dependence in an approximate manner. We fix the problem for Adam by explicitly normalizing each weight vector, and by optimizing only its direction, such that the effective learning rate can be precisely controlled.

- We further show that, without proper regularization, the learning signal backpropagated from the softmax layer may vary with the overall magnitude of the logits in an undesirable way. Based on the observation, we apply batch normalization or L2-regularization to the logits, which further improves the generalization performance in classification tasks.

In essence, our proposed methods, ND-Adam and regularized softmax, improve the generalization performance of Adam by enabling more precise control over the directions of parameter updates, the learning rates, and the learning signals.

## 2 BACKGROUND AND MOTIVATION

### 2.1 ADAPTIVE MOMENT ESTIMATION (ADAM)

Adaptive moment estimation (Adam) (Kingma & Ba, 2014) is a stochastic optimization method that applies individual adaptive learning rates to different parameters, based on the estimates of the first and second moments of the gradients. Specifically, for $n$ trainable parameters, $\theta \in \mathbb{R}^n$, Adam maintains a running average of the first and second moments of the gradient w.r.t. each parameter as

$$m_t = \beta_1 m_{t-1} + (1 - \beta_1) g_t, \tag{1a}$$

$$v_t = \beta_2 v_{t-1} + (1 - \beta_2) g_t^2. \tag{1b}$$

Here, $t$ denotes the time step, $m_t \in \mathbb{R}^n$ and $v_t \in \mathbb{R}^n$ denote respectively the first and second moments, and $\beta_1 \in \mathbb{R}$ and $\beta_2 \in \mathbb{R}$ are the corresponding decay factors. Kingma & Ba (2014) further notice that, since $m_0$ and $v_0$ are initialized to 0's, they are biased towards zero during the initial time steps, especially when the decay factors are large (i.e., close to 1). Thus, for computing the next update, they need to be corrected as

$$\hat{m}_t = \frac{m_t}{1 - \beta_1^t}, \hat{v}_t = \frac{v_t}{1 - \beta_2^t}, \tag{2}$$

where $\beta_1^t$, $\beta_2^t$ are the $t$-th powers of $\beta_1$, $\beta_2$ respectively. Then, we can update each parameter as

$$\theta_t = \theta_{t-1} - \frac{\alpha_t}{\sqrt{\hat{v}_t} + \epsilon} \hat{m}_t, \tag{3}$$

where $\alpha_t$ is the global learning rate, and $\epsilon$ is a small constant to avoid division by zero. Note the above computations between vectors are element-wise.

A distinguishing merit of Adam is that the magnitudes of parameter updates are invariant to rescaling of the gradient, as shown by the adaptive learning rate term, $\frac{\alpha_t}{\sqrt{\hat{v}_t} + \epsilon}$. However, there are two potential problems when applying Adam to DNNs.

First, in some scenarios, DNNs trained with Adam generalize worse than that trained with stochastic gradient descent (SGD) (Wilson et al., 2017). Zhang et al. (2017) demonstrate that over-parameterized DNNs are capable of memorizing the entire dataset, no matter if it is natural data

or meaningless noise data, and thus suggest much of the generalization power of DNNs comes from the training algorithm, e.g., SGD and its variants. It coincides with another recent work (Wilson et al., 2017), which shows that simple SGD often yields better generalization performance than adaptive gradient methods, such as Adam. As pointed out by the latter, the difference in the generalization performance may result from the different directions of updates. Specifically, for each hidden unit, the SGD update of its input weight vector can only lie in the span of all possible input vectors, which, however, is not the case for Adam due to the individually adapted learning rates. We refer to this problem as the *direction missing problem*.

Second, while batch normalization (Ioffe & Szegedy, 2015) can significantly accelerate the convergence of DNNs, the input weights and the scaling factor of each hidden unit can be scaled in infinitely many (but consistent) ways, without changing the function implemented by the hidden unit. Thus, for different magnitudes of an input weight vector, the updates given by Adam can have different effects on the overall network function, which is undesirable. Furthermore, even when batch normalization is not used, a network using linear rectifiers (e.g., ReLU, leaky ReLU) as activation functions, is still subject to ill-conditioning of the parameterization (Glorot et al., 2011), and hence the same problem. We refer to this problem as the *ill-conditioning problem*.

## 2.2 L2 WEIGHT DECAY

L2 weight decay is a regularization technique frequently used with SGD. It often has a significant effect on the generalization performance of DNNs. Despite the simplicity and crucial role of L2 weight decay in the training process, it remains to be explained how it works in DNNs. A common justification for L2 weight decay is that it can be introduced by placing a Gaussian prior upon the weights, when the objective is to find the maximum a posteriori (MAP) weights (Blundell et al., 2015). However, as discussed in Sec. 2.1, the magnitudes of input weight vectors are irrelevant in terms of the overall network function, in some common scenarios, rendering the variance of the Gaussian prior meaningless.

We propose to view L2 weight decay in neural networks as a form of weight normalization, which may better explain its effect on the generalization performance. Consider a neural network trained with the following loss function:

$$\widetilde{L}\left(\theta;\mathcal{D}\right) = L\left(\theta;\mathcal{D}\right) + \frac{\lambda}{2} \sum_{i \in \mathcal{N}} \|w_i\|_2^2, \tag{4}$$

where $L\left(\theta;\mathcal{D}\right)$ is the original loss function specified by the task, $\mathcal{D}$ is a batch of training data, $\mathcal{N}$ is the set of all hidden units, and $w_i$ denotes the input weights of hidden unit $i$, which is included in the trainable parameters, $\theta$. For simplicity, we consider SGD updates without momentum. Therefore, the update of $w_i$ at each time step is

$$\Delta w_i = -\alpha \frac{\partial \widetilde{L}}{\partial w_i} = -\alpha \left( \frac{\partial L}{\partial w_i} + \lambda w_i \right), \tag{5}$$

where $\alpha$ is the learning rate. As we can see from Eq. (5), the gradient magnitude of the L2 penalty is proportional to $\|w_i\|_2$, thus forms a negative feedback loop that stabilizes $\|w_i\|_2$ to an equilibrium value. Empirically, we find that $\|w_i\|_2$ tends to increase or decrease dramatically at the beginning of the training, and then varies mildly within a small range, which indicates $\|w_i\|_2 \approx \|w_i + \Delta w_i\|_2$. In practice, we usually have $\|\Delta w_i\|_2 / \|w_i\|_2 \ll 1$, thus $\Delta w_i$ is approximately orthogonal to $w_i$, i.e. $w_i \cdot \Delta w_i \approx 0$.

Let $l_{\|w_i}$ and $l_{\perp w_i}$ be the vector projection and rejection of $\frac{\partial L}{\partial w_i}$ on $w_i$, which are defined as

$$l_{\|w_i} = \left( \frac{\partial L}{\partial w_i} \cdot \frac{w_i}{\|w_i\|_2} \right) \frac{w_i}{\|w_i\|_2}, l_{\perp w_i} = \frac{\partial L}{\partial w_i} - l_{\|w_i}. \tag{6}$$

From Eq. (5) and (6), it is easy to show

$$\frac{\|\Delta w_i\|_2}{\|w_i\|_2} \approx \frac{\|l_{\perp w_i}\|_2}{\|l_{\|w_i}\|_2} \alpha \lambda. \tag{7}$$

As discussed in Sec. 2.1, when batch normalization is used, or when linear rectifiers are used as activation functions, the magnitude of $\|w_i\|_2$ is irrelevant. Thus, it is the direction of $w_i$ that actually

makes a difference in the overall network function. If L2 weight decay is not applied, the magnitude of $w_i$'s direction change will decrease as $\|w_i\|_2$ increases during the training process, which can potentially lead to overfitting (discussed in detail in Sec. 3.2). On the other hand, Eq. (7) shows that L2 weight decay implicitly normalizes the weights, such that the magnitude of $w_i$'s direction change does not depend on $\|w_i\|_2$, and can be tuned by the product of $\alpha$ and $\lambda$. In the following, we refer to $\|\Delta w_i\|_2 / \|w_i\|_2$ as the *effective learning rate* of $w_i$.

While L2 weight decay produces the normalization effect in an implicit and approximate way, we will show that explicitly doing so enables more precise control of the effective learning rate.

## 3 Normalized Direction-preserving Adam

We first present the normalized direction-preserving Adam (ND-Adam) algorithm, which essentially improves the optimization of the input weights of hidden units, while employing the vanilla Adam algorithm to update other parameters. Specifically, we divide the trainable parameters, $\theta$, into two sets, $\theta^v$ and $\theta^s$, such that $\theta^v = \{w_i | i \in \mathcal{N}\}$, and $\theta^s = \{\theta \setminus \theta^v\}$. Then we update $\theta^v$ and $\theta^s$ by different rules, as described by Alg. 1. The learning rates for the two sets of parameters are denoted respectively by $\alpha_t^v$ and $\alpha_t^s$.

---

**Algorithm 1:** Normalized direction-preserving Adam

```
/* Initialization                                                        */
```
$t \leftarrow 0$;
**for** $i \in \mathcal{N}$ **do**
    $w_{i,0} \leftarrow w_{i,0} / \|w_{i,0}\|_2$;
    $m_0(w_i) \leftarrow 0$;
    $v_0(w_i) \leftarrow 0$;
```
/* Perform T iterations of training                                      */
```
**while** $t < T$ **do**
    $t \leftarrow t + 1$;
```
    /* Update θ^v                                                     */
```
    **for** $i \in \mathcal{N}$ **do**
        $\bar{g}_t(w_i) \leftarrow \partial L / \partial w_i$;
        $g_t(w_i) \leftarrow \bar{g}_t(w_i) - (\bar{g}_t(w_i) \cdot w_{i,t-1}) w_{i,t-1}$;
        $m_t(w_i) \leftarrow \beta_1 m_{t-1}(w_i) + (1 - \beta_1) g_t(w_i)$;
        $v_t(w_i) \leftarrow \beta_2 v_{t-1}(w_i) + (1 - \beta_2) \|g_t(w_i)\|_2^2$;
        $\hat{m}_t(w_i) \leftarrow m_t(w_i) / (1 - \beta_1^t)$;
        $\hat{v}_t(w_i) \leftarrow v_t(w_i) / (1 - \beta_2^t)$;
        $\bar{w}_{i,t} \leftarrow w_{i,t-1} - \alpha_t^v \hat{m}_t(w_i) / \left( \sqrt{\hat{v}_t(w_i)} + \epsilon \right)$;
        $w_{i,t} \leftarrow \bar{w}_{i,t} / \|\bar{w}_{i,t}\|_2$;
```
    /* Update θ^s using Adam                                          */
```
    $\theta_t^s \leftarrow \text{AdamUpdate}\left(\theta_{t-1}^s; \alpha_t^s, \beta_1, \beta_2\right)$;
**return** $\theta_T$;

---

In Alg. 1, the iteration over $\mathcal{N}$ can be performed in parallel, and thus introduces no extra computational complexity. Compared to Adam, computing $g_t(w_i)$ and $w_{i,t}$ may take slightly more time, which, however, is negligible in practice. On the other hand, to estimate the second order moment of each $w_i \in \mathbb{R}^n$, Adam maintains $n$ scalars, whereas ND-Adam requires only one scalar, $v_t(w_i)$. Thus, ND-Adam has smaller memory overhead than Adam.

In the following, we address the direction missing problem and the ill-conditioning problem discussed in Sec. 2.1, and explain Alg. 1 in detail. We show how the proposed algorithm jointly solves the two problems, as well as its relation to other normalization schemes.

### 3.1 PRESERVING GRADIENT DIRECTIONS

Assuming the stationarity of a hidden unit's input distribution, the SGD update (possibly with momentum) of the input weight vector is a linear combination of historical gradients, and thus can only lie in the span of the input vectors. As a result, the input weight vector itself will eventually converge to the same subspace.

On the contrary, the Adam algorithm adapts the global learning rate to each scalar parameter independently, such that the gradient of each parameter is normalized by a running average of its magnitudes, which changes the direction of the gradient. To preserve the direction of the gradient w.r.t. each input weight vector, we generalize the learning rate adaptation scheme from scalars to vectors.

Let $g_t(w_i)$, $m_t(w_i)$, $v_t(w_i)$ be the counterparts of $g_t$, $m_t$, $v_t$ for vector $w_i$. Since Eq. (1a) is a linear combination of historical gradients, it can be extended to vectors without any change; or equivalently, we can rewrite it for each vector as

$$m_t(w_i) = \beta_1 m_{t-1}(w_i) + (1 - \beta_1) g_t(w_i). \tag{8}$$

We then extend Eq. (1b) as

$$v_t(w_i) = \beta_2 v_{t-1}(w_i) + (1 - \beta_2) \|g_t(w_i)\|_2^2, \tag{9}$$

i.e., instead of estimating the average gradient magnitude for each individual parameter, we estimate the average of $\|g_t(w_i)\|_2^2$ for each vector $w_i$. In addition, we modify Eq. (2) and (3) accordingly as

$$\hat{m}_t(w_i) = \frac{m_t(w_i)}{1 - \beta_1^t}, \hat{v}_t(w_i) = \frac{v_t(w_i)}{1 - \beta_2^t}, \tag{10}$$

and

$$w_{i,t} = w_{i,t-1} - \frac{\alpha_t^v}{\sqrt{\hat{v}_t(w_i)} + \epsilon} \hat{m}_t(w_i). \tag{11}$$

Here, $\hat{m}_t(w_i)$ is a vector with the same dimension as $w_i$, whereas $\hat{v}_t(w_i)$ is a scalar. Therefore, when applying Eq. (11), the direction of the update is the negative direction of $\hat{m}_t(w_i)$, and thus is in the span of the historical gradients of $w_i$.

It is worth noting that only the input to the first layer (i.e., the training data) is stationary throughout training. Thus, for the weights of an upper layer to converge to the span of its input vectors, it is necessary for the lower layers to converge first. Interestingly, this predicted phenomenon may have been observed in practice (Brock et al., 2017).

Despite the empirical success of SGD, a question remains as to why it is desirable to constrain the input weights in the span of the input vectors. A possible explanation is related to the manifold hypothesis, which suggests that real-world data presented in high dimensional spaces (images, audios, text, etc) concentrates on manifolds of much lower dimensionality (Cayton, 2005; Narayanan & Mitter, 2010). In fact, commonly used activation functions, such as (leaky) ReLU, sigmoid, tanh, can only be activated (not saturating or having small gradients) by a portion of the input vectors, in whose span the input weights lie upon convergence. Assuming the local linearity of the manifolds of data or hidden-layer representations, constraining the input weights in the subspace that contains some of the input vectors, encourages the hidden units to form local coordinate systems on the corresponding manifold, which can lead to good representations (Rifai et al., 2011).

### 3.2 SPHERICAL WEIGHT OPTIMIZATION

The ill-conditioning problem occurs when the magnitude change of an input weight vector can be compensated by other parameters, such as the scaling factor of batch normalization, or the output weight vector, without affecting the overall network function. Consequently, suppose we have two DNNs that parameterize the same function, but with some of the input weight vectors having different magnitudes, applying the same SGD or Adam update rule will, in general, change the network functions in different ways. Thus, the ill-conditioning problem makes the training process inconsistent and difficult to control.

More importantly, when the weights are not properly regularized (e.g., without using L2 weight decay), the magnitude of $w_i$'s direction change will decrease as $\|w_i\|_2$ increases during the training

process. As a result, the effective learning rate for $w_i$ tends to decrease faster than expected, making the network converge to sharp minima. It is well known that sharp minima generalize worse than flat minima (Hochreiter & Schmidhuber, 1997; Keskar et al., 2017).

As shown in Sec. 2.2, L2 weight decay can alleviate the ill-conditioning problem by implicitly and approximately normalizing the weights. However, we still do not have a precise control over the effective learning rate, since $\|l_{\perp w_i}\|_2 / \|l_{\|w_i}\|_2$ is unknown and not necessarily stable. Moreover, the approximation fails when $\|w_i\|_2$ is far from the equilibrium due to improper initialization, or drastic changes in the magnitudes of the weight vectors. This problem is also addressed by (Neyshabur et al., 2015), by employing a geometry invariant to rescaling of weights. However, their proposed methods do not preserve the direction of gradient.

To address the ill-conditioning problem in a more principled way, we restrict the L2-norm of each $w_i$ to 1, and only optimize its direction. In other words, instead of optimizing $w_i$ in a $n$-dimensional space, we optimize $w_i$ on a $(n-1)$-dimensional unit sphere. Specifically, we first obtain the raw gradient w.r.t. $w_i$, $\bar{g}_t(w_i) = \partial L / \partial w_i$, and project the gradient onto the unit sphere as

$$g_t(w_i) = \bar{g}_t(w_i) - (\bar{g}_t(w_i) \cdot w_{i,t-1}) w_{i,t-1}. \tag{12}$$

Here, $\|w_{i,t-1}\|_2 = 1$. Then we follow Eq. (8)-(10), and replace (11) with

$$\bar{w}_{i,t} = w_{i,t-1} - \frac{\alpha_t^v}{\sqrt{\hat{v}_t(w_i)} + \epsilon} \hat{m}_t(w_i), \tag{13a}$$

and

$$w_{i,t} = \frac{\bar{w}_{i,t}}{\|\bar{w}_{i,t}\|_2}. \tag{13b}$$

In Eq. (12), we keep only the component that is orthogonal to $w_{i,t-1}$. However, $\hat{m}_t(w_i)$ is not necessarily orthogonal as well. In addition, even when $\hat{m}_t(w_i)$ is orthogonal to $w_{i,t-1}$, Eq. (13a) can still increase $\|w_i\|_2$, according to the Pythagorean theorem. Therefore, we explicitly normalize $w_{i,t}$ in Eq. (13b), to ensure $\|w_{i,t}\|_2 = 1$ after each update. Also note that, since $w_{i,t-1}$ is a linear combination of its historical gradients, $g_t(w_i)$ still lies in the span of the historical gradients after the projection in Eq. (12).

Compared to SGD with L2 weight decay, spherical weight optimization explicitly normalizes the weight vectors, such that each update to the weight vectors only changes their directions, and strictly keeps the magnitudes constant. As a result, the effective learning rate of a weight vector is

$$\frac{\|\Delta w_{i,t}\|_2}{\|w_{i,t-1}\|_2} \approx \frac{\|\hat{m}_t(w_i)\|_2}{\sqrt{\hat{v}_t(w_i)}} \alpha_t^v, \tag{14}$$

which enables precise control over the learning rate of $w_i$ through a single hyperparameter, $\alpha_t^v$, rather than two as required by Eq. (7). Note that it is possible to control the effective learning rate more precisely, by normalizing $\hat{m}_t(w_i)$ with $\|\hat{m}_t(w_i)\|_2$, instead of by $\sqrt{\hat{v}_t(w_i)}$. However, by doing so, we lose the information provided by $\|\hat{m}_t(w_i)\|_2$ at different time steps. In addition, since $\hat{m}_t(w_i)$ is less noisy than $g_t(w_i)$, $\|\hat{m}_t(w_i)\|_2 / \sqrt{\hat{v}_t(w_i)}$ becomes small near convergence, which is considered a desirable property of Adam (Kingma & Ba, 2014). Thus, we keep the gradient normalization scheme intact.

We note the difference between various gradient normalization schemes and the normalization scheme employed by spherical weight optimization. As shown in Eq. 11, ND-Adam generalizes the gradient normalization scheme of Adam, and thus both Adam and ND-Adam normalize the gradient by a running average of its magnitude. This, and other similar schemes (Hazan et al., 2015; Yu et al., 2017) make the optimization less susceptible to vanishing and exploding gradients. On the other hand, the proposed spherical weight optimization serves a different purpose. It normalizes each weight vector and projects the gradient onto a unit sphere, such that the effective learning rate can be controlled more precisely. Moreover, it provides robustness to improper weight initialization, since the magnitude of each weight vector is kept constant.

For nonlinear activation functions, such as sigmoid and tanh, an extra scaling factor is needed for each hidden unit to express functions that require unnormalized weight vectors. For instance, given an input vector $x \in \mathbb{R}^n$, and a nonlinearity $\phi(\cdot)$, the activation of hidden unit $i$ is then given by

$$y_i = \phi(\gamma_i w_i \cdot x + b_i), \tag{15}$$

where $\gamma_i$ is the scaling factor, and $b_i$ is the bias.

### 3.3 Relation to Weight Normalization and Batch Normalization

A related normalization and reparameterization scheme, weight normalization (Salimans & Kingma, 2016), has been developed as an alternative to batch normalization, aiming to accelerate the convergence of SGD optimization. We note the difference between spherical weight optimization and weight normalization. First, the weight vector of each hidden unit is not directly normalized in weight normalization, i.e, $\|w_i\|_2 \neq 1$ in general. At training time, the activation of hidden unit $i$ is

$$y_i = \phi \left( \frac{\gamma_i}{\|w_i\|_2} w_i \cdot x + b_i \right), \tag{16}$$

which is equivalent to Eq. (15) for the forward pass. For the backward pass, the effective learning rate still depends on $\|w_i\|_2$ in weight normalization, hence it does not solve the ill-conditioning problem. At inference time, both of these two schemes can combine $w_i$ and $\gamma_i$ into a single equivalent weight vector, $w'_i = \gamma_i w_i$, or $w'_i = \frac{\gamma_i}{\|w_i\|_2} w_i$.

While spherical weight optimization naturally encompasses weight normalization, it can further benefit from batch normalization. When combined with batch normalization, Eq. (15) evolves into

$$y_i = \phi \left( \gamma_i \, \mathrm{BN} \left( w_i \cdot x \right) + b_i \right), \tag{17}$$

where $\mathrm{BN} \left( \cdot \right)$ represents the transformation done by batch normalization without scaling and shifting. Here, $\gamma_i$ serves as the scaling factor for both the normalized weight vector and batch normalization. At training time, the distribution of the input vector, $x$, changes over time, slowing down the training of the sub-network composed by the upper layers. Salimans & Kingma (2016) observe that, such problem cannot be eliminated by normalizing the weight vectors alone, but can be substantially mitigated by combining weight normalization and mean-only batch normalization.

Additionally, in linear rectifier networks, the scaling factors, $\gamma_i$, can be removed (or set to 1), without changing the overall network function. Since $w_i \cdot x$ is standardized by batch normalization, we have

$$\mathbb{E}_x \left[ \mathrm{BN} \left( w_i \cdot x \right)^2 \right] \approx 1, \tag{18}$$

and hence

$$\mathrm{Var}_x \left[ \mathrm{BN} \left( w_i \cdot x \right) + b_i \right] \approx 1. \tag{19}$$

Therefore, $y_i$'s that belong to the same layer, or different dimensions of $x$ that fed to the upper layer, will also have comparable variances, which potentially makes the weight updates of the upper layer more stable. For these reasons, we combine the use of spherical weight optimization and batch normalization, as shown in Eq. (17).

## 4 Regularized Softmax

For multi-class classification tasks, the softmax function is the *de facto* activation function for the output layer. Despite its simplicity and intuitive probabilistic interpretation, we observe a related problem to the ill-conditioning problem we have addressed. Similar to how different magnitudes of weight vectors result in different updates to the same network function, the learning signal back-propagated from the softmax layer varies with the overall magnitude of the logits.

Specifically, when using cross entropy as the surrogate loss with one-hot target vectors, the prediction is considered correct as long as $\arg\max_{c \in \mathcal{C}} (z_c)$ is the target class, where $z_c$ is the logit before the softmax activation, corresponding to category $c \in \mathcal{C}$. Thus, the logits can be positively scaled together without changing the predictions, whereas the cross entropy and its derivatives will vary with the scaling factor. Concretely, denoting the scaling factor by $\eta$, the gradient w.r.t. each logit is

$$\frac{\partial L}{\partial z_{\hat{c}}} = \eta \left[ \frac{\exp \left( \eta z_{\hat{c}} \right)}{\sum_{c \in \mathcal{C}} \exp \left( \eta z_c \right)} - 1 \right], \tag{20a}$$

and

$$\frac{\partial L}{\partial z_{\bar{c}}} = \frac{\eta \exp \left( \eta z_{\bar{c}} \right)}{\sum_{c \in \mathcal{C}} \exp \left( \eta z_c \right)}. \tag{20b}$$

where $\hat{c}$ is the target class, and $\bar{c} \in \mathcal{C} \backslash \{\hat{c}\}$.

For Adam and ND-Adam, since the gradient w.r.t. each scalar or vector are normalized, the absolute magnitudes of Eq. (20a) and (20b) are irrelevant. Instead, the relative magnitudes make a difference here. When $\eta$ is small, we have

$$\lim_{\eta \to 0} \left| \frac{\partial L/\partial z_{\bar{c}}}{\partial L/\partial z_{\hat{c}}} \right| = \frac{1}{|\mathcal{C}| - 1}, \tag{21}$$

which indicates that, when the magnitude of the logits is small, softmax encourages the logit of the target class to increase, while equally penalizing that of the other classes. On the other end of the spectrum, assuming no two digits are the same, we have

$$\lim_{\eta \to \infty} \left| \frac{\partial L/\partial z_{\bar{c}'}}{\partial L/\partial z_{\hat{c}}} \right| = 1, \ \lim_{\eta \to \infty} \left| \frac{\partial L/\partial z_{\bar{c}''}}{\partial L/\partial z_{\hat{c}}} \right| = 0, \tag{22}$$

where $\bar{c}' = \arg\max_{c \in \mathcal{C} \setminus \{\hat{c}\}} (z_c)$, and $\bar{c}'' \in \mathcal{C} \setminus \{\hat{c}, \bar{c}'\}$. Eq. (22) indicates that, when the magnitude of the logits is large, softmax penalizes only the largest logit of the non-target classes. The latter case is related to the saturation problem of softmax discussed in Oland et al. (2017). However, they focus on the problem of small absolute gradient magnitude, which does not affect Adam and ND-Adam.

It is worth noting that both of these two cases can happen without the scaling factor. For instance, varying the norm of the weights of the softmax layer is equivalent to varying the value of $\eta$, in terms of the relative magnitude of the gradient. In the case of small $\eta$, the logits of all non-target classes are penalized equally, regardless of the difference in $\hat{z} - \bar{z}$ for different $\bar{z} \in \mathcal{C} \setminus \{\hat{z}\}$. However, it is more reasonable to penalize more the logits that are closer to $\hat{z}$, which are more likely to cause mis-classification. In the case of large $\eta$, although the logit that is most likely to cause misclassification is strongly penalized, the logits of other non-target classes are ignored. As a result, the logits of the non-target classes tend to be similar at convergence, ignoring the fact that some classes are closer to each other than the others.

We propose two methods to exploit the prior knowledge that the magnitude of the logits should not be too small or too large. First, we can apply batch normalization to the logits. But instead of setting $\gamma_c$'s as trainable variables, we consider them as a single hyperparameter, $\gamma_{\mathcal{C}}$, such that $\gamma_c = \gamma_{\mathcal{C}}, \forall c \in \mathcal{C}$. Tuning the value of $\gamma_{\mathcal{C}}$ can lead to a better trade-off between the two extremes described by Eq. (21) and (22). The optimal value of $\gamma_{\mathcal{C}}$ tends to remain the same for different optimizers or different network widths, but varies with dataset and network depth. We refer to this method as batch-normalized softmax (BN-Softmax).

Alternatively, since the magnitude of the logits tends to grow larger than expected (in order to minimize the cross entropy), we can apply L2-regularization to the logits by adding the following penalty to the loss function:

$$L_{\mathcal{C}} = \frac{\lambda_{\mathcal{C}}}{2} \sum_{c \in \mathcal{C}} z_c^2, \tag{23}$$

where $\lambda_{\mathcal{C}}$ is a hyperparameter to be tuned. Different from BN-Softmax, $\lambda_{\mathcal{C}}$ can be shared by different datasets and networks of different depths.

## 5 EXPERIMENTS

In this section, we provide empirical evidence for the analysis in Sec. 2.2, and evaluate the performance of ND-Adam and regularized softmax on CIFAR-10 and CIFAR-100.

### 5.1 THE EFFECT OF L2 WEIGHT DECAY

To empirically examine the effect of L2 weight decay, we train a wide residual network (WRN) (Zagoruyko & Komodakis, 2016b) of 22 layers, with a width of 7.5 times that of a vanilla ResNet. Using the notation in Zagoruyko & Komodakis (2016b), we refer to this network as WRN-22-7.5. We train the network on the CIFAR-10 dataset (Krizhevsky & Hinton, 2009), with a small modification to the original WRN architecture, and with a different learning rate annealing schedule. Specifically, for simplicity and slightly better performance, we replace the last fully connected layer with a convolutional layer with 10 output feature maps. I.e., we change the layers after the last residual block from `BN-ReLU-GlobalAvgPool-FC-Softmax` to

`BN-ReLU-Conv-GlobalAvgPool-Softmax`. In addition, for clearer comparisons, the learning rate is annealed according to a cosine function without restart (Loshchilov & Hutter, 2016; Gastaldi, 2017). We train the model for 80k iterations with a batch size of 128, similar to the settings in Zagoruyko & Komodakis (2016b). The experiments are based on a TensorFlow implementation of WRN (Wu, 2016).

As a common practice, we use SGD with a momentum of 0.9, the analysis for which is similar to that in Sec. 2.2. Due to the linearity of derivatives and momentum, $\Delta w_i$ can be decomposed as $\Delta w_i = \Delta w_i^l + \Delta w_i^p$, where $\Delta w_i^l$ and $\Delta w_i^p$ are the components corresponding to the original loss function, $L(\cdot)$, and the L2 penalty term (see Eq. (4)), respectively. Fig. 1a shows the ratio between the scalar projection of $\Delta w_i^l$ on $\Delta w_i^p$ and $\|\Delta w_i^p\|_2$, which indicates how the tendency of $\Delta w_i^l$ to increase $\|w_i\|_2$ is compensated by $\Delta w_i^p$. Note that $\Delta w_i^p$ points to the negative direction of $w_i$, even when momentum is used, since the direction change of $w_i$ is slow. As shown in Fig. 1a, at the beginning of the training, $\Delta w_i^p$ dominants and quickly adjusts $\|w_i\|_2$ to its equilibrium value. During the middle stage of the training, the projection of $\Delta w_i^l$ on $\Delta w_i^p$, and $\Delta w_i^p$ almost cancel each other out. Then, near the end of the training, the gradient of $w_i$ diminishes rapidly to near zero, making $\Delta w_i^p$ dominant again. Therefore, Eq. (7) holds more accurately during the middle stage of the training.

In Fig. 1b, we show how the effective learning rate varies in different hyperparameter settings. By Eq. (7), $\|\Delta w_i\|_2 / \|w_i\|_2$ is expected to remain the same as long as $\alpha\lambda$ stays constant, which is confirmed by the fact that the curve for $\alpha_0 = 0.1, \lambda = 0.001$ overlaps with that for $\alpha_0 = 0.05, \lambda = 0.002$. However, comparing the curve for $\alpha_0 = 0.1, \lambda = 0.001$, with that for $\alpha_0 = 0.1, \lambda = 0.0005$, we can see that the value of $\|\Delta w_i\|_2 / \|w_i\|_2$ does not change proportionally to $\alpha\lambda$. On the other hand, by using ND-Adam, we can control the value of $\|\Delta w_i\|_2 / \|w_i\|_2$ more precisely by adjusting the learning rate for weight vectors, $\alpha^v$. For the same training step, changes in $\alpha^v$ lead to approximately proportional changes in $\|\Delta w_i\|_2 / \|w_i\|_2$, as shown by the two curves corresponding to ND-Adam in Fig. 1b.

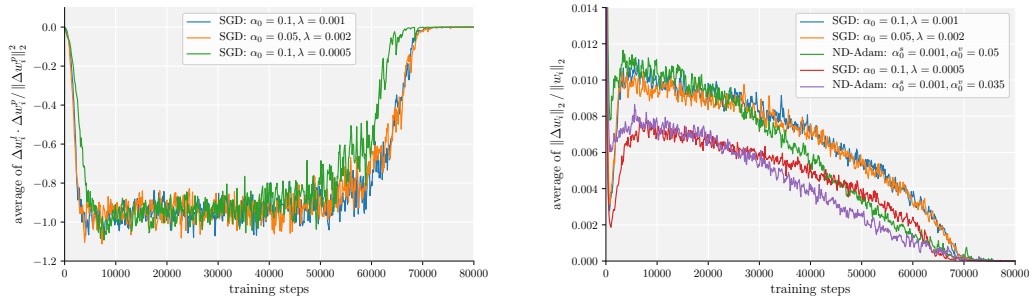

(a) The scalar projection of $\Delta w_i^l$ on $\Delta w_i^p$ normalized by $\|\Delta w_i^p\|_2$.

(b) The relative magnitude of the weight updates, or the effective learning rate.

Figure 1: An illustration of how L2 weight decay and ND-Adam control the effective learning rate. The results are obtained from the 5th layer of the network, and other layers show similar results.

## 5.2 PERFORMANCE EVALUATION

To compare the generalization performance of SGD, Adam, and ND-Adam, we train the same WRN-22-7.5 network on the CIFAR-10 and CIFAR-100 datasets. For SGD and ND-Adam, we first tune the hyperparameters for SGD ($\alpha_0 = 0.1, \lambda = 0.001$, momentum 0.9), then tune the initial learning rate of ND-Adam for weight vectors to match the effective learning rate to that of SGD ($\alpha_0^v = 0.05$), as shown in Fig. 1b. While L2 weight decay can greatly affect the performance of SGD, it does not noticeably benefit Adam in our experiments. For Adam and ND-Adam, $\beta_1$ and $\beta_2$ are set to the default values of Adam, i.e., $\beta_1 = 0.9, \beta_2 = 0.999$. Although the learning rate of Adam is usually set to a constant value, we observe better performance with the cosine learning rate schedule. The initial learning rate of Adam ($\alpha_0$), and that of ND-Adam for scalar parameters ($\alpha_0^s$) are both tuned

to 0.001. We use the same data augmentation scheme as used in Zagoruyko & Komodakis (2016b), including horizontal flips and random crops, but no dropout is used.

We first experiment with the use of trainable scaling parameters ($\gamma_i$) of batch normalization. As shown in Fig. 2b, at convergence, the test accuracies of ND-Adam are significantly improved upon that of vanilla Adam, and matches that of SGD. Note that at the early stage of training, the training losses of Adam drop dramatically as shown in Fig. 2a, and the test accuracies also increase more rapidly than that of ND-Adam and SGD. However, the test accuracies remain at a high level afterwards, which indicates that Adam tends to quickly find and get stuck in bad local minima that do not generalize well.

The average results of 3 runs are summarized in the first part of Table 1. Interestingly, compared to SGD, ND-Adam shows slightly better performance on CIFAR-10, but worse performance on CIFAR-100. This inconsistency may be related to the problem of softmax discussed in Sec. 4, that there is a lack of proper control over the magnitude of the logits. But overall, given comparable effective learning rates, ND-Adam and SGD show similar generalization performance. In this sense, the effective learning rate is a more natural learning rate measure than the learning rate hyperparameters.

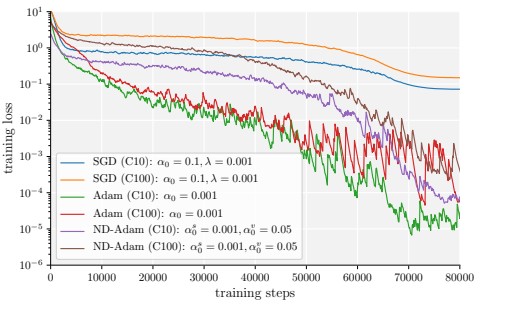 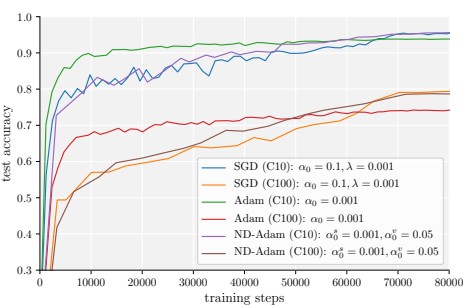

(a) The training losses on CIFAR-10/100.      (b) The test accuracies on CIFAR-10/100.

Figure 2: The training losses and test accuracies of the same network trained with SGD, Adam, and ND-Adam. Batch normalization with scaling factors is used.

Next, we repeat the experiments with the use of BN-Softmax. As discussed in Sec. 3.2, $\gamma_i$'s can be removed from a linear rectifier network, without changing the overall network function. Although this property does not strictly hold for residual networks due to the skip connections, we find that simply removing the scaling factors results in slightly improved generalization performance when using ND-Adam. However, the improvement is not consistent as it degrades performance of SGD. Interestingly, when BN-Softmax is further used, we observe consistent improvement over all three algorithms. Thus, we only report results for this setting. The scaling factor of the logits, $\gamma_{\mathcal{C}}$, is set to 2.5 for CIFAR-10, and 1 for CIFAR-100. As shown in the second part of Table 1, BN-Softmax significantly improves the performance of Adam and ND-Adam. Moreover, in this setting, we obtain the best generalization performance with ND-Adam, outperforming SGD and Adam on both CIFAR-10 and CIFAR-100.

While the TensorFlow implementation we use already provides an adequate test bed, we notice that it is different from the original implementation of WRN in several aspects. For instance, they use different nonlinearities (leaky ReLU vs. ReLU), and use different skip connections for downsampling (average pooling vs. strided convolution). A seemingly subtle but important difference is that, L2-regularization is applied not only to weight vectors, but also to the scales and biases of batch normalization in the original implementation, which leads to better generalization performance. For further comparison between SGD and ND-Adam, we reimplement ND-Adam and test its performance on a PyTorch version of the original implementation (Zagoruyko & Komodakis, 2016a).

Due to the aforementioned differences, we use a slightly different hyperparameter setting in this experiment. Specifically, for SGD $\lambda$ is set to 5e$-$4, while for ND-Adam $\lambda$ is set to 5e$-$6 (L2-regularization for biases), and both $\alpha_0^s$ and $\alpha_0^v$ are set to 0.04. In this case, regularizing softmax

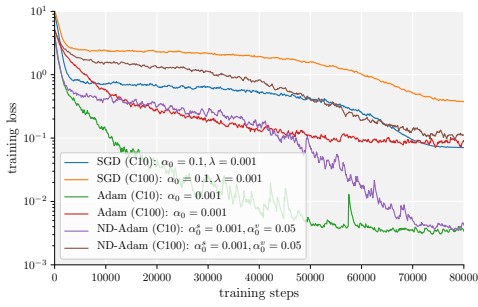

(a) The training losses on CIFAR-10/100.

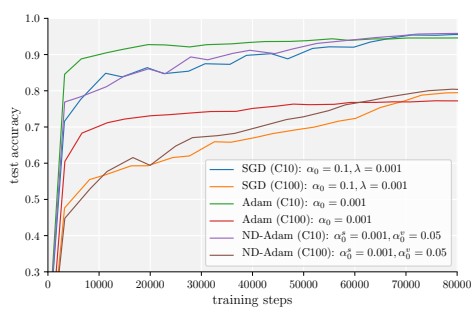

(b) The test accuracies on CIFAR-10/100.

Figure 3: The training losses and test accuracies of the same network trained with SGD, Adam, and ND-Adam. Batch normalization *without* scaling factors, and BN-Softmax are used.

| Method | CIFAR-10 Error (%) | CIFAR-100 Error (%) |
|---|---|---|
| BN w/ scaling factors | | |
| SGD | 4.61 | 20.60 |
| Adam | 6.14 | 25.51 |
| ND-Adam | 4.53 | 21.45 |
| BN w/o scaling factors, BN-Softmax | | |
| SGD | 4.49 | 20.18 |
| Adam | 5.43 | 22.48 |
| ND-Adam | **4.14** | **19.90** |

Table 1: Test error rates of WRN-22-7.5 networks on CIFAR-10 and CIFAR-100. Based on a TensorFlow implementation of WRN.

does not yield improved performance for SGD, since the L2-regularization applied to $\gamma_i$'s and the last layer weights can serve a similar purpose. Thus, we only apply L2-regularized softmax for ND-Adam with $\lambda_{\mathcal{C}} = 0.001$. The average results of 3 runs are summarized in Table 2. Note that the performance of SGD for WRN-28-10 is slightly better than that reported with the original implementation (i.e., $4.00$ and $19.25$), due to the modifications described in Sec. 5.1. In this experiment, SGD and ND-Adam show almost identical generalization performance.

| Method | CIFAR-10 Error (%) | CIFAR-100 Error (%) |
|---|---|---|
| WRN-22-7.5 | | |
| SGD | 3.84 | **19.24** |
| ND-Adam | **3.70** | 19.30 |
| WRN-28-10 | | |
| SGD | 3.80 | 18.48 |
| ND-Adam | **3.70** | **18.42** |

Table 2: Test error rates of WRN-22-7.5 and WRN-28-10 networks on CIFAR-10 and CIFAR-100. Based on the original implementation of WRN.

## 6    CONCLUSION

In this paper, we have introduced ND-Adam, a tailored version of Adam for training DNNs, to bridge the generalization gap between Adam and SGD. ND-Adam is designed to preserve the direction of gradient for each weight vector, and produce the regularization effect of L2 weight decay in a more precise and principled way. Moreover, we have introduced regularized softmax, which limits the magnitude of softmax logits to provide better learning signals. By combining ND-Adam and regularized softmax, our experiments have shown significantly improved generalization performance, eliminating the gap between Adam and SGD. From a high-level view, our analysis and empirical results suggest the need for more precise control over the training process of DNNs.

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
