# OpenReview forum: "Normalized Direction-preserving Adam"
_ICLR.cc/2018/Conference — Reject_

### Official Review · AnonReviewer2 · 2017-11-27
**Some related works should be analyzed. The experimental validation should to be revised.**

**Rating:** 5
**Confidence:** 4

**Review:**

Method:

The paper is missing analysis of some important related works such as

"Beyond convexity: Stochastic quasi-convex optimization" by E. Hazan et al. (2015)

where Stochastic Normalized Gradient Descent (SNGD) was proposed.

Then, normalized gradient versions of AdaGrad and Adam were proposed in

"Normalized Gradient with Adaptive Stepsize Method for Deep
Neural Network Training" by A. W. Yu et al. (2017).

Another work which I find to be relevant is

"Follow the Signs for Robust Stochastic Optimization" by L. Balles and P. Hennig (2017).

From my personal experiments, restricting w_i to have L2 norm of 1, i.e., to be +-1
leads to worse generalization. One reason for this is that weight decay is not
really functioning since it cannot move w_i to 0 or make its amplitude any smaller.
Please correct me if I misunderstand something here.

The presence of +-1 weights moves us to the area of low-precision NNs,
or more specifically, NNs with binary / binarized weights as in

"BinaryConnect: Training Deep Neural Networks with
binary weights during propagations" by M. Courbariaux et al. (2015)

and

"Binarized neural networks: Training deep neural networks with weights and activations constrained to+ 1 or-1" by M. Courbariaux et al. (2016).

Regarding
"Moreover, the magnitude of each update does not depend on themagnitude of the gradient. Thus, ND-Adam is more robust to improper initialization, and vanishing or exploding gradients."

If the magnitude of each update does not depend on the magnitude of the gradient, then the algorithm heavily depends on the learning rate. Otherwise, it does not have any means to approach the optimum in a reasonable number of steps *when* it is initialized very / unreasonably far from it. The claim of your second sentence is not supported by the paper.

Evaluation:

I am not confident that the presented experimental validation is fair. First, the original WRN paper and many other papers with ResNets used weight decay of 0.0005 and not 0.001 or 0.002 as used for SGD in this paper. It is unclear why this setting was changed. One could just use \alpha_0 = 0.05 and \lambda = 0.0005.

Then, I don't see why the authors use WRN-22-7.5 which is different from WRN-28-10 which was suggested in the original study and used in several follow-up works. The difference between WRN-22-7.5 and WRN-28-10 is unlikely to be significant,
the former might have about only 2 times less parameters which should barely change the final validation errors. However, the use of WRN-22-7.5 makes it impossible to easily compare the presented results to the results of Zagoruyko who had 3.8\% with WRN-28-10. I believe that the use of the setup of Zagoruyko for WRN-22-7.5 would allow to get much better results than 4.5\% and 4.49\% shown for SGD and likely better 4.14\% shown for ND-Adam. I note that the use of WRN-22-7.5 is unlikely to be due to the used hardware because later in paper the authors refer to WRN-34-7.5.

My intuition is that the proposed ND-Adam moves the algorithm back to SGD but with potentially harmful constraints of w_i=+-1. Even the values of \alpha^v_0 found for ND-Adam (e.g., \alpha^v_0=0.05 in Figure 1B) are in line of what would be optimal values of \alpha_0 for SGD.

I find it uncomfortable that BN-Softmax is introduced here to support the use of an optimization algorithm, moreover, that the values of \gamma_c are different for CIFAR-10 and CIFAR-100. I wonder if the proposed values are optimal (and therefore selected) for all three tested algorithms  or only for Adam-ND. I expect that hyperparameters of SGD and Adam would also need to be revised to account for BN-Softmax.

---

> ### Author Response · Authors · 2017-12-14
> **Responses**
>
> Dear reviewer,
>
> Thank you for your comments.
>
> 1. The paper is missing analysis of some important related works. From my personal experiments, restricting w_i to have L2 norm of 1, i.e., to be +-1 leads to worse generalization. One reason for this is that weight decay is not really functioning since it cannot move w_i to 0 or make its amplitude any smaller. Please correct me if I misunderstand something here. The presence of +-1 weights moves us to the area of low-precision NNs.
>
> Thank you for pointing out the related works on normalized gradient, we will discuss about them in the paper. We note here the difference between the normalization done by ND-Adam and gradient normalization. ND-Adam normalizes the gradient by a running average of its magnitude, but this is just something inherited from Adam. In ND-Adam, we further normalize the input weight vector of each hidden unit, as a more precise way to produce the normalization effect of L2 weight decay (see Eq. (7) and (14)).
>
> We agree with your point that restricting w_i to have L2 norm of 1 can lead to worse generalization. And the reason is exactly as you stated, L2 weight decay does not work on the restricted weights. Our analysis in Sec 2.2 shows that L2 weight decay implicitly normalizes weight vectors, and thus keeps the effective learning rate (defined as |\delta w|/|w| in Sec. 2.2) from decreasing undesirably. For SGD, restricting weight norms to 1 eliminates the normalization effect and leads to smaller effective learning rate than expected. For ND-Adam, on the other hand, the effective learning rate can be controlled in a more precise way (see Eq. (14)) with normalized weight vectors, as we replace L2 weight decay with the proposed spherical weight optimization.
>
> By normalizing a weight vector, we only restrict its norm, rather than the individual weights. So each weight can have any real value between -1 and 1 (as long as the vector norm is 1), instead of just +-1. A trainable scaling factor can be multiplied to the weight vector in case other norm values than 1 are necessary, and the scaling factor of batch normalization can be shared for this purpose (see Sec. 3.3). Therefore, normalizing weight vectors does not reduce the expressiveness of the model. On the other hand, the expressiveness of low-precision NNs, such as binary networks, are often significantly weakened as a trade-off for consuming less computational resources.
>
> 2. "Moreover, the magnitude of each update does not depend on the magnitude of the gradient. Thus, ND-Adam is more robust to improper initialization, and vanishing or exploding gradients." If the magnitude of each update does not depend on the magnitude of the gradient, then the algorithm heavily depends on the learning rate. Otherwise, it does not have any means to approach the optimum in a reasonable number of steps *when* it is initialized very / unreasonably far from it. The claim of your second sentence is not supported by the paper.
>
> The first sentence of the quote is not clear enough, and thank you for pointing it out. For Adam and ND-Adam, each parameter update is normalized by a running average of the gradient magnitude, thus is less susceptible to small/large gradient magnitude than SGD. In many cases, such as this one, both Adam/ND-Adam and SGD work best with a decaying learning rate rather than a constant one. Indeed, Adam with a constant learning rate is sometimes preferred over SGD due to better optimization performance, such as for GAN training. However, as clarified above, it is a property inherited from Adam. For ND-Adam, we emphasize the technique we propose (i.e., spherical weight optimization) to optimize the directions of weight vectors, which normalizes each weight vector and projects the gradient onto a unit sphere, such that the effective learning rate can be controlled more precisely. Moreover, since we only optimize the direction of each weight vector and keep its magnitude constant, the initial magnitude doesn't matter for ND-Adam as it does for SGD. We will explain it more clearly in the paper.

---

> ### Author Response · Authors · 2017-12-14
> **Responses (cont. 1)**
>
> 3. I am not confident that the presented experimental validation is fair. First, the original WRN paper and many other papers with ResNets used weight decay of 0.0005 and not 0.001 or 0.002 as used for SGD in this paper. It is unclear why this setting was changed. One could just use \alpha_0 = 0.05 and \lambda = 0.0005.
>
> Our experiments were based on a TensorFlow implementation of WRN available at https://github.com/tensorflow/models/tree/master/research/resnet. We have recently compared this implementation to a PyTorch implementation provided by the WRN paper at https://github.com/szagoruyko/wide-residual-networks, and found the following differences:
> +--------------------------------------+-------------------------+-----------------------------------------------+
> |                                                  | TensorFlow impl. | PyTorch impl.                                    |
> +--------------------------------------+-------------------------+-----------------------------------------------+
> | Input standardization          | Per sample           | Per dataset                                        |
> +--------------------------------------+-------------------------+-----------------------------------------------+
> | Skip conn. between stages | Avg. pooling         | 1 by 1 conv.                                       |
> +--------------------------------------+-------------------------+-----------------------------------------------+
> | Nonlinearity                           | Leaky ReLU (0.1)  | ReLU                                                  |
> +--------------------------------------+-------------------------+-----------------------------------------------+
> | L2 regularization                   | Weights only        | Weights, scales and biases of BN |
> +--------------------------------------+-------------------------+-----------------------------------------------+
>
> There are other subtle differences that are not listed here, such as different parameter initializations. Due to these differences, the two implementations may have different optimal hyperparameter configurations. In the PyTorch implementation, they use the initial learning rate and weight decay configuration of (0.1, 0.0005) with a multi-step learning rate schedule. In our preliminary experiments, we find the configuration of (0.1, 0.001) slightly better than (0.1, 0.0005) with the cosine learning rate schedule we use, but the results are very similar. We use the configuration of (0.05, 0.002) only to show that the effective learning rate of weights stays the same as long as the product of learning rate and weight decay stays the same (see Eq. (7) and Fig.1(b)), though the performance is also very close to that of (0.1, 0.001) due to the same effective learning rate.

---

> ### Author Response · Authors · 2017-12-14
> **Responses (cont. 2)**
>
> 4. Then, I don't see why the authors use WRN-22-7.5 which is different from WRN-28-10 which was suggested in the original study and used in several follow-up works. The difference between WRN-22-7.5 and WRN-28-10 is unlikely to be significant, the former might have about only 2 times less parameters which should barely change the final validation errors. However, the use of WRN-22-7.5 makes it impossible to easily compare the presented results to the results of Zagoruyko who had 3.8\% with WRN-28-10. I believe that the use of the setup of Zagoruyko for WRN-22-7.5 would allow to get much better results than 4.5\% and 4.49\% shown for SGD and likely better 4.14\% shown for ND-Adam. I note that the use of WRN-22-7.5 is unlikely to be due to the used hardware because later in paper the authors refer to WRN-34-7.5.
>
> We use WRN-22-7.5 instead of WRN-28-10 due to limited hardware resources. We have tested WRN-28-10 on CIFAR-10, which was more than two times slower than WRN-22-7.5 (the latter took about 10 hours on a single GTX 1080 GPU), but only slightly outperforms WRN-22-7.5 (as you mentioned) for both SGD and ND-Adam. To obtain the results in Table 1, we need 36 runs in total, so we decide to use WRN-22-7.5.
>
> Due to the differences mentioned above, the generalization performance of the TensorFlow implementation is worse than that of the original WRN implementation. As you can find in the GitHub links above, the error rates of WRN-28-10 on CIFAR-10 are 5% and 4% (the 3.8% you mentioned is the result of WRN-40-10 with dropout), respectively. The performance of SGD reported in our paper is actually improved upon the original TensorFlow implementation, by using a cosine learning rate schedule and by removing the FC layer (see Sec. 5.1). Since we are not trying to improve WRN itself, we believe the TensorFlow implementation of WRN is a fairly good testbed for comparing the performance of SGD, Adam, and ND-Adam.
>
> We would like to emphasize that the main contribution of our work is to identify why Adam generalizes worse than SGD, and give a solution to fix the problem. However, we agree with you that comparing with the original implementation of WRN would be more convincing. So we have reimplemented ND-Adam with PyTorch. Interestingly, we find that applying L2 regularization on the scales and biases of BN is indeed important for achieving better generalization performance. However, in this case SGD does not benefit from explicitly regularizing softmax, which is likely due to the L2 regularization applied on BN scales, as it indirectly restricts the magnitude of the softmax logits. For fair comparison, we also apply L2 regularization to the biases when using ND-Adam (the scales are not used by ND-Adam). The average results of 2 runs are summarized in the following table:
> +----------------+-------------------+--------------------+
> |                     |  WRN-22-7.5  |   WRN-28-10   |
> +                     +-------------------+--------------------+
> |                     |  C10  |  C100  |  C10  |  C100  |
> +----------------+--------+----------+--------+----------+
> | SGD (orig.) | 4.15 |     ---     | 4.00 |  19.25  |
> +----------------+--------+----------+--------+----------+
> | SGD             | 3.84 |  19.24  | 3.80  |  18.37  |
> +----------------+--------+----------+--------+----------+
> | ND-Adam   | 3.70 |  19.24  | 3.68  |  18.37  |
> +----------------+--------+----------+--------+----------+
> For both SGD and ND-Adam, we have slightly modified the implementation as done before, i.e., we use a cosine learning rate schedule and remove the FC layer. Some results of the original WRN implementation (SGD (orig.)) are also included for comparison. We will update the paper and the code accordingly.
>
> 5. My intuition is that the proposed ND-Adam moves the algorithm back to SGD but with potentially harmful constraints of w_i=+-1. Even the values of \alpha^v_0 found for ND-Adam (e.g., \alpha^v_0=0.05 in Figure 1B) are in line of what would be optimal values of \alpha_0 for SGD.
>
> We normalize each weight vector to have a norm of 1, but do not constrain each weight to be +-1. As stated in the response to the first comment, the normalization does not reduce the expressiveness of the model. As you observed in Fig. 1(b), we deliberately match the effective learning rate of ND-Adam to that of SGD. And by showing that similar effective learning rates lead to similar generalization performance, we argue that the effective learning rate is a more natural learning rate measure than the learning rate hyperparameters (Sec. 5.2).

---

> ### Author Response · Authors · 2017-12-14
> **Responses (cont. 3)**
>
> 6. I find it uncomfortable that BN-Softmax is introduced here to support the use of an optimization algorithm, moreover, that the values of \gamma_c are different for CIFAR-10 and CIFAR-100. I wonder if the proposed values are optimal (and therefore selected) for all three tested algorithms  or only for Adam-ND. I expect that hyperparameters of SGD and Adam would also need to be revised to account for BN-Softmax.
>
> One of the problems that degrades the generalization performance of Adam (or SGD without L2 weight decay) is that, for different magnitudes of an input weight vector, the updates given by the same update rule can have different effects on the overall network function. For example, if we increase the magnitude of a weight vector without changing the overall network function, the same update rule will result in a smaller effective learning rate for this weight vector. While this problem can be alleviated by L2 weight decay or solved by the proposed spherical weight optimization, a similar problem also exists for the softmax layer. As discussed in Sec. 4, we can scale the logits by a positive factor without changing the predictions of the model, whereas the gradient backpropagated from it can vary greatly, which motivates the idea of batch-normalized softmax. Regularizing softmax is important for fixing the problem we have identified. We will explain more on this motivation in the paper.
>
> Batch-normalized softmax is a simple way to constrain the magnitude of the logits. In our experiments, \gamma_c is chosen from {1, 1.5, 2.5, ...}, and was tuned for SGD and ND-Adam. SGD and ND-Adam share the same optimal values of \gamma_c, which also significantly improve the performance of Adam. In addition, we have recently simplified BN-Softmax to an L2 penalty on the logits, the value of which can be shared by different dataset and models with different layers. But unlike BN-Softmax, the value of the L2 penalty is not likely to be shared by SGD or Adam, since the same penalty may lead to different magnitudes of the logits. The experimental results are shown in the table above. We will describe the details in the paper.

---

### Official Review · AnonReviewer1 · 2017-11-27
**A variant of ADAM optimization algorithm that normalizes the weights of each hidden unit**

**Rating:** 5
**Confidence:** 5

**Review:**

This paper proposes a variant of ADAM optimization algorithm that normalizes the weights of each hidden unit. They further suggest using batch normalization on the output of the network before softmax to improve the generalization. The main ideas are new to me and the paper is well-written. The arguments and derivations are very clear. However, the experimental results suggest that the proposed method is not superior to SGD and ADAM.

Pros:

- The idea of optimizing the direction while ignoring the magnitude is interesting and make sense.
- Using batch normalization before softmax is interesting.

Cons:

- In the abstract, authors claim that the proposed method has good optimization performance of ADAM and good generalization performance of SGD. Such a method could be helpful if one can get to the same level of generalization faster (less number of epochs). However, the experiments suggest that optimization advantages of the proposed method do not translate to faster generalization. Figures 2,3 and Table 1 indicate that the generalization performance of this method is very similar to SGD.

- The paper is not coherent. In particular, direction-preserving ADAM and batch-normalized softmax trick are completely orthogonal ideas.

- In the introduction and Section 2.2, authors claim that weight decay has a significant effect on the generalization performance of DNNs. I wonder if authors can refer to any work on this. My own experience and several empirical works have suggested that weight decay does not improve generalization significantly.

---

> ### Author Response · Authors · 2017-12-14
> **Responses**
>
> Dear reviewer,
>
> Thank you for your comments.
>
> 1. In the abstract, authors claim that the proposed method has good optimization performance of ADAM and good generalization performance of SGD. Such a method could be helpful if one can get to the same level of generalization faster (less number of epochs). However, the experiments suggest that optimization advantages of the proposed method do not translate to faster generalization. Figures 2,3 and Table 1 indicate that the generalization performance of this method is very similar to SGD.
>
> Thank you for pointing this out, and we agree with you that better generalization performance should help get to the same level of generalization faster. In this case (i.e., training a wide residual network on CIFAR-10/100), optimization is not difficult for either SGD or ND-Adam, but we expect the advantage to become more significant when optimization is difficult. For example, Adam is often preferred over SGD when training GANs, due to better optimization performance.
>
> However, since the potential advantage in optimization is not strongly supported by the experiments in this case, we will rephrase this part of the paper. Instead, we would like to emphasize that the main contribution of our work is to identify why Adam generalizes worse than SGD, and give a solution to fix the problem. The hyperparameters of ND-Adam are not extensively tuned, but are tuned to match the effective learning rate (defined as |\delta w|/|w| in Sec. 2.2) of SGD, as shown in Fig. 1b. Nevertheless, we show that the generalization performance is significantly improved upon that of Adam, and also outperforms SGD when batch-normalized softmax is used.
>
> 2. The paper is not coherent. In particular, direction-preserving ADAM and batch-normalized softmax trick are completely orthogonal ideas.
>
> One of the problems that degrades the generalization performance of Adam (or SGD without L2 weight decay) is that, for different magnitudes of an input weight vector, the updates given by the same update rule can have different effects on the overall network function. For example, if we increase the magnitude of a weight vector without changing the overall network function, the same update rule will result in a smaller effective learning rate for this weight vector. While this problem can be alleviated by L2 weight decay or solved by the proposed spherical weight optimization, a similar problem also exists for the softmax layer. As discussed in Sec. 4, we can scale the logits by a positive factor without changing the predictions of the model, whereas the gradient backpropagated from it can vary greatly, which motivates the idea of batch-normalized softmax. Regularizing softmax is important for fixing the problem we have identified. We will explain more on this motivation in the paper.
>
> 3. In the introduction and Section 2.2, authors claim that weight decay has a significant effect on the generalization performance of DNNs. I wonder if authors can refer to any work on this. My own experience and several empirical works have suggested that weight decay does not improve generalization significantly.
>
> L2 weight decay is widely used in state-of-the-art models for image classification, examples include residual networks (He et al., 2016), Inception/Xception (Chollet, 2016), and squeeze-and-excitation networks (Hu et al., 2017). To our knowledge, it is also used in language models, although it is not always stated in papers (e.g., see the code at https://github.com/zihangdai/mos). In this work, we find that a major function of L2 weight decay is to implicitly normalize weight vectors (see Eq. (7)), in order to keep the effective learning rate from decreasing undesirably. As a result, L2 weight decay may not be very useful when decreasing learning rate is not a problem.

---

### Official Review · AnonReviewer3 · 2017-11-28

**Rating:** 4
**Confidence:** 4

**Review:**

The paper extended the Adam optimization algorithm to preserve the update direction. Instead of using the un-centered variance of individual weights, the proposed method adapts the learning rate for the incoming weights to a hidden unit jointly using the L2 norm of the gradient vector. The authors empirically demonstrated the method works well on CIFAR-10/100 tasks.

Comments:

- I found the paper very hard to follow. The authors could improve the clarity of the paper greatly by listing their contribution clearly for readers to digest. The authors also combined the proposed method with a few existing deep learning tricks in the paper. All those tricks that, ie. section 3.3 and 4, should go into the background section.

- Overall, the only contribution of the paper seems to be the ad-hoc modification to Adam in Eq. (9). Why is this a reasonable modification? Do we expect this modification to fail in any circumstances? The experiments on CIFAR dataset and one CNN architecture do not provide enough evidence to show the proposed method work well in general.

---

> ### Author Response · Authors · 2017-12-14
> **Responses**
>
> Dear reviewer,
>
> Thank you for your comments.
>
> 1. I found the paper very hard to follow. The authors could improve the clarity of the paper greatly by listing their contribution clearly for readers to digest. The authors also combined the proposed method with a few existing deep learning tricks in the paper. All those tricks that, ie. section 3.3 and 4, should go into the background section.
>
> Thanks for the suggestion, we will list our contribution in the introduction. We also clarify the contribution as follows. We first identify two problems that degrade the generalization performance of Adam (Sec. 2): 1) the direction of Adam update does not lie in the span of historical gradients, which can lead to drastically different solutions than SGD in some cases (Wilson et al., 2017), and 2) the effective learning rates (defined as |\delta w|/|w| in Sec. 2.2) of both Adam and SGD tend to decrease as the norms of weight vectors increase during training, and lead to sharp local minima that do not generalize well. We further show that, when combined with SGD, L2 weight decay can implicitly and approximately normalize weight vectors, such that the effective learning rate is no longer dependent on the norms of weight vectors (see Eq. (7)). The normalization view of L2 weight decay provides a more concrete explanation for how it works for DNNs. Next, we fix the first problem by adapting the learning rate to each weight vector, instead of each individual weight, such that the direction of the gradient is preserved. We fix the second problem by explicitly normalizing each weight vector, which produces the normalization effect of L2 weight decay in a more precise way (see Eq. (14)). Finally, we propose batch-normalized softmax to regularize the learning signal backpropagated from the softmax layer.
>
> Sec. 3.3 explains the relationship between the proposed spherical weight optimization (Sec. 3.2) and batch/weight normalization. For instance, we show that the scaling factors of BN can be shared with ND-Adam as the scales of weight vectors. To our knowledge, BN-Softmax described in Sec. 4 is a simple but new idea, and it works very well for Adam and ND-Adam.
>
> 2. Overall, the only contribution of the paper seems to be the ad-hoc modification to Adam in Eq. (9). Why is this a reasonable modification? Do we expect this modification to fail in any circumstances? The experiments on CIFAR dataset and one CNN architecture do not provide enough evidence to show the proposed method work well in general.
>
> The modifications to Adam are in Eq. (9), (12) and (13b). Here we explain the rationale behind these modifications. The updates of SGD lie in the span of historical gradients, whereas it is not the case for Adam. This difference between Adam (as well as other adaptive gradient methods) and SGD has been discussed in Wilson et al., 2017, where they show it can lead to drastically different but worse solutions compared to SGD. Eq. (9) eliminates this difference, while keeping the adaptive learning rate scheme intact for weight vectors. Eq. (12) projects the gradient of each weight vector onto a unit sphere, and Eq. (13b) ensures the weight vector stays on the unit sphere. This modification is designed to distill the normalization effect of L2 weight decay and apply it to Adam. The latter problem is also addressed by a concurrent work submitted to ICLR 2018 (https://openreview.net/forum?id=rk6qdGgCZ).

---

> > ### Comment · AnonReviewer3 · 2018-01-16
> >
> > Thank the authors for their response. I am disappointed the latest revised paper does not provide any further insight into the ND-Adam updates. It is with much regret that my score remains the same.

---

### Public Comment · (anonymous) · 2017-11-27
**Related work**

There is probably a missing related work: https://arxiv.org/pdf/1707.04822.pdf

In that work, the gradient is normalized to preserve the direction, while the adaptive step size (Adam) is used. This sounds exactly match your paper title. Could you elaborate the difference if there is any?

---

> ### Author Response · Authors · 2017-11-27
> **Responses**
>
> Thank you for your comments.
>
> In https://arxiv.org/pdf/1707.04822.pdf, they normalize the gradient of each step to keep only its direction. However, the gradient is further multiplied by individually adapted step sizes to form the actual updates, thus changing the direction of the gradient.
>
> In this work, we normalized the input weight vector of each hidden unit, rather than the gradient, at the end of each step. We also adapt the learning rate to weight vectors instead of individual weights, in order to preserve the direction of gradient.

---

### Author Response · Authors · 2017-11-27
**Code**

Code can be found at https://github.com/zj10/ND-Adam.

---

### Author Response · Authors · 2018-01-04
**Paper update**

Dear reviewers,

Thank you very much for your valuable comments. We have updated the paper according to the comments. The major changes are listed below:

1. We list our contribution in the introduction.

2. In Sec. 3.2, we note the difference between various gradient normalization schemes and the normalization
scheme employed ND-Adam. We also briefly explain how they improve the robustness to vanishing/exploding gradients, and to improper weight initialization.

3. In Sec. 4, we explain more about the rationale behind regularized softmax. We show that softmax exhibits a similar problem to one of the problems we have identified in Adam. Moreover, as an alternative to batch-normalized softmax, we propose to apply L2-regularization to the softmax logits, which serves the same purpose as batch-normalized softmax, but is easier to use.

4. Due to several differences, the performance of the original WRN implementation is better than that of the TensorFlow implementation we use. Therefore, in Sec. 5, we further compare the performance of SGD and ND-Adam based on the original WRN implementation. The results confirm that ND-Adam eliminates the generalization gap between Adam and SGD. The code for the experiments is available at https://github.com/zj10/ND-Adam.

---

### Decision · Program_Chairs · 2018-01-29
**ICLR 2018 Conference Acceptance Decision**

**Decision:**

Reject

**Comment:**

The paper proposes a modification to Adam which is intended to ensure that the direction of weight update lies in the span of the historical gradients and to ensure that the effective learning rate does not decrease as the magnitudes of the weights increase.  The reviewers wanted a clearer justification of the changes made to Adam and a more extensive evaluation, and held to this opinion after reading the authors' rebuttal and revisions.

Pros:
+ The basic idea of treating the direction and magnitude separately in the optimization is interesting.

Cons:
- Insufficient evaluation of the new method.
- More justification and analysis needed for the modifications.  For example, are there circumstances under which they will fail?
- The modification to Adam and batch-normalized softmax idea are orthogonal to one another, making for a less coherent story.
- Proposed method does not have better generalization performance than SGD.
- Concern that constraining weight vectors to the unit sphere can harm generalization.